# Data-Driven Analysis of Risk-Assessment Methods for Cold Food Chains

**DOI:** 10.3390/foods12081677

**Published:** 2023-04-17

**Authors:** Qian Wang, Zhiyao Zhao, Zhaoyang Wang

**Affiliations:** 1College of Artificial Intelligence, Beijing Technology and Business University, Beijing 100048, China; 2China Food Flavor and Nutrition Health Innovation Center, Beijing Technology and Business University, Beijing 100048, China

**Keywords:** cold-chain food safety, risk assessment, data-driven model, knowledge graph

## Abstract

The problem of cold-chain food safety is becoming increasingly prominent. Cold food chain risk assessment is an important way to ensure cold-chain food safety. Using CiteSpace, this study analyzes the knowledge map of research hotspots in the field of cold-chain food safety over the past 18 years, identifies the research keywords, presents the centrality statistics, and calculates the cluster values and average cluster contour values. Adopting a data-driven perspective, risk-assessment methods for cold food chains are summarized based on qualitative risk assessment, quantitative risk assessment, and comprehensive qualitative and quantitative risk assessment. The advantages and disadvantages of each are summarized. Finally, the problems and challenges in current cold food chain risk-assessment research are summarized in three aspects: the data credibility of cold food chain traceability systems, cold-chain food safety audit methods, and nontraditional cold food chain risk assessment. Suggestions are given for strengthening the cold food chain risk-assessment system to provide a decision-making reference to help regulatory authorities take risk prevention and control measures.

## 1. Introduction

With increasing types and quantities of food, food safety issues are becoming increasingly prominent, attracting attention worldwide [1]. According to the World Health Organization, approximately 600 million people contract foodborne diseases every year [2]. In China alone, on average, more than 200 million people suffer from foodborne diseases every year [3]. Foodborne diseases pose a serious threat to public health. Aside from grain, fresh food products are a main source of daily nutrition, and demand is constantly growing. However, fresh products can easily deteriorate and breed bacteria, thus causing food safety problems [4]. The fresh food supply chain requires low temperatures throughout the process to inhibit the growth of harmful microorganisms. In this regard, cold-chain logistics has become an effective means to ensure the quality and safety of fresh products. Cold chains have higher requirements regarding timeliness, vulnerability, and safety, as well as higher risk coefficients. Therefore, the risk assessment of cold food chains is an important way to ensure cold-chain food safety [5,6,7].

Food risk assessment refers to the use of relevant methods to assess factors and risks that may have adverse effects on humans. It is mainly divided into four parts: hazard identification, hazard description, exposure assessment, and risk description [8]. The data-driven method regards the system as a black-box model and does not need to consider complex mechanism processes. The system model is fitted by a mathematical model trained on input and output data, effectively avoiding the complexity and uncertainty of the modeling process. With the integration of information technology and data, data-driven methods have been applied to food risk assessment, becoming a research hotspot in the field of food safety [9,10,11]. A cold food chain risk assessment model based on food monitoring data can scientifically analyze current levels of cold-chain food risk, predict development trends in cold-chain food risk, and achieve the cross-integration of cold-chain food safety and data-driven approaches.

Using the document visualization analysis function in CiteSpace, this study analyzes the knowledge map of research hotspots regarding data-driven methods in the field of cold food chain safety. It outlines cold food chain risk assessment in terms of cold food chain risk assessment, introduces different data-driven methods in cold food chain risk assessment, and summarizes the advantages and disadvantages of different data-driven approaches. The problems and challenges of the current data-driven approach in the field of cold food chain risk assessment are presented to provide theoretical support and a scientific reference for the development of cold food chain risk assessment.

## 2. Research Hotspot Analysis

### 2.1. Data Collection

The research data in this section uses the China National Knowledge Internet (CNKI) as the data source. CNKI is an internationally leading online publishing platform that aims to achieve the dissemination, sharing, and value-added utilization of knowledge resources throughout society, integrating academic journals, papers, conferences, newspapers, books, yearbooks, patents, standards, achievements, and academic journals. When using CNKI for literature retrieval, we chose advanced search for the retrieval mode, with “cold chain,” “fresh products,” “food safety,” “risk,” and “risk assessment” as the theme or keywords. The literature source category selected academic journals and degree papers, and the time range was set to 2006–2023. After manual screening, 120 valid documents for analysis were finally obtained. The selected documents were exported in RefWorks format, then CiteSpace software (6.1.R6) was used to perform format conversion and visual analysis on text data, drawing a map of scientific knowledge of relevant documents.

### 2.2. Research Hotspot Analysis

The co-occurrence and clustering functions in the CiteSpace keyword analysis were used to draw a map of scientific knowledge to further analyze research hotspots in the field of food safety. Keywords in the food safety field from 2006–2023 were extracted and a keyword co-occurrence map drawn (Figure 1). There are 213 nodes and 481 connections in the figure, with a density of 0.0213. The size of a node represents the size of the keyword co-occurrence frequency. The larger the node, the higher the node’s word frequency, the greater the number of co-words, and the greater the centrality in the network. The thickness of the line reflects the co-occurrence relationship between different keywords. The thicker the line, the closer the likelihood of keyword co-occurrence. We calculated the centrality of the keywords, and screened out the top 20 keywords with high frequency and centrality. A table was created showing the research keywords and centrality statistics (Table 1). The frequency of keyword citations and centrality are positively correlated with the degree of attention received. The keywords of high frequency and high-mindedness represent research hotspots in this field. We can see in Table 1 that the keywords with high centrality are “cold-chain logistics,” “risk assessment,” “fresh agricultural product,” “Bayesian network (BN),” and “back-propagation (BP) neural networks.” These show that data-driven research on food safety, risk assessment, and prediction plays an important role in the cold food chain field and has been a research hotspot in recent years. Keyword clustering analysis is then performed on the keyword co-occurrence network to obtain a keyword clustering analysis knowledge map with eight main clustering keywords, as shown in Figure 2. The total module clustering value is Q = 0.6732, which is greater than 0.3; this indicates a significant clustering effect. The average clustering contour value is S = 0.9093, which is greater than 0.5; this indicates that the analysis result is reliable. As can be seen from Figure 2, Cluster # 0 and Cluster # 1 indicate that the research field of food cold chain mainly focuses on cold-chain logistics and fresh agricultural products. Cluster # 2, Cluster # 3, and Cluster # 5 indicate that the research direction of the food cold chain mainly focuses on three aspects: risk assessment, food safety, and risk assessment. Cluster # 4, Cluster # 6, and Cluster # 7 indicate that food cold-chain research methods mainly focus on the Internet of Things, BP neural network, and analytic hierarchy process (AHP)—fuzzy comprehensive evaluation. Based on the research keywords and year of their first appearance in the central statistical table, it can be seen that in recent years, the food cold chain has shown a development trend of research direction, research field, and research methods. Specifically, in 2011, high-frequency keywords such as risk assessment, risk assessment, and food safety first appeared in the field of food cold chain, which means that food safety and risk have become the mainstream research direction of the food cold chain. Based on this, diversified risk research directions such as risk management, risk identification, risk control, influencing factors, and supply chain risk were gradually expanded in 2014 and 2016. In 2012, the field of food cold chain was first subdivided into specific research areas such as cold-chain logistics, fresh agricultural product, and supply chain. Scholars began to conduct research in different directions in specific research areas, achieving the integration of research fields and research directions. From 2013 to 2018, data-driven methods such as fuzzy comprehensive evaluation, BP neural network, BN, fuzzy hierarchical comprehensive evaluation, AHP and the Delphi method were gradually applied to different research fields and research directions. Since then, the food cold chain has achieved the vertical development and horizontal integration of research fields, research directions, and research methods.

## 3. Overview of Cold Food Chain Risk Assessment

A cold food chain is a type of supply chain that keeps perishable fresh foods, such as fruits and vegetables, dairy products, meat, eggs, and fish, in a low-temperature environment in the production, storage, transportation, and sales links to maintain food quality and safety, reduce food waste, and prevent pollution. The process of cold food chain risk assessment can be roughly divided into four stages: hazard identification, hazard description, exposure assessment, and risk description. Hazard identification is the process of identifying risk factors (physical, biological, and chemical) that could have adverse effects on human health in the cold food chain [12]. Hazard description refers to describing the possible adverse effects, severity, and tolerance of risk factors in humans; this mainly involves examining dose–response relationships to qualitatively or quantitatively evaluate the hazard process of risk factors for humans [13,14]. Exposure assessment estimates the possibility of an individual’s exposure to the hazards of risk factors and the possible intake [15]. Since the number of hazards in a cold chain dynamically changes, it is difficult for risk-assessment personnel to accurately predict the number of pathogenic bacteria in the food before consumption; it is necessary, therefore, to use a model to predict the number of risk factors for the human body. Risk description is a qualitative or quantitative estimate of the possibility and severity of adverse effects from risk factors based on information from the hazard identification, hazard description, and exposure assessment; it also describes the uncertainty of risk assessment itself [16].

## 4. Application of a Data-Driven Model for Cold Food Chain Risk Assessment

Data-driven food risk assessment is a process of grading and assessing food risks according to the risk prediction results of a data-driven model. This is an important part of risk description. According to the results of the data-driven model, food risk assessment can be divided into qualitative risk assessment, quantitative risk assessment, and comprehensive qualitative and quantitative risk assessment.

### 4.1. Qualitative Risk Assessment

Qualitative risk assessment is a method based on experience, knowledge, and expert questionnaire data. Risks are analyzed and judged based on risk assessors’ experience and knowledge, and the survey data, and then the risk results are described [17]. Qualitative risk assessment does not need to establish an accurate quantitative model, which avoids the difficulties of modeling and can intuitively observe risk results. Qualitative risk-assessment methods are mainly divided into the Delphi method and decision-tree analysis. The Delphi method, also known as the expert survey method, uses back-to-back communication following certain procedures to solicit the opinions of members of an expert group, conduct repeated questionnaires, and form consensus by listening to the opinions of all parties [18,19]. Using the Delphi method, Zhou et al. [20] solicited the opinions of 16 experts in relevant fields. On that basis, they obtained qualitative and quantitative data for different levels of hazard probability regarding pathogenic microorganism combinations related to 188 livestock and poultry products as an early-warning threshold for risk assessment. Xu [21], meanwhile, used two rounds of Delphi correspondence questionnaires distributed to experts. The results were used to determine three first-level indicators, nine second-level indicators, and 32 third-level indicators for logistics risk, supply risk, and platform risk. On that basis, an evaluation index system was built for the quality and safety risk of fresh products on e-commerce platforms. Wang et al. [22] took into account the scientific and subjective characteristics of the cold-chain logistics demand forecast index of fresh agricultural products. They used the Delphi method for three rounds of consultation and finally constructed a cold-chain logistics demand forecast index system for fresh agricultural products. Kim et al. [23] used three rounds of Delphi technology screening for food safety goals through 15 experts, resulting in a consensus rate of at least 70% for 58 of the 75 food safety goals, providing guidance for the evaluation of refrigerated food safety levels in retail delicatessen stores.

Meanwhile, decision-tree analysis is a common method in data mining classification algorithms. It is represented as a tree structure; a leaf node represents a conclusion, an internal node defines an attribute, and a path from top to bottom determines the classification rule structure. By building a decision-tree model, the risk of monitored products can be divided according to classification rules [24]. Using a decision-tree model, Zhao et al. [25] divided food sampled in Shandong Province, China, into two categories—high risk and low risk—according to the classification of food and the difference in specific unqualified food additive items. Sheng et al. [26] established three decision trees to sort pollutants, food additives, and biological toxins in noodles and divided the monitored substances into high, medium, and low priority.

In food risk assessment, the Delphi method is normally used to screen risk indicators, classify risk indicators into hazard levels, and then build a risk assessment indicator system. Decision-tree analysis is widely used in the risk classification and risk priority identification of foodborne hazards. This is conducive to determining the selection order of hazards in food risk assessment and can provide a reference for food risk assessment. Qualitative risk-assessment methods are mostly used in the risk identification stage. They mainly rely on the subjective judgments of evaluators and a logical understanding of causal relationships. The assessment process is simple. Although the risk results can be directly observed, there are also some problems. First, qualitative risk-assessment methods rely too much on the subjective judgment of experts, lack quantitative analysis, and cannot quantify risks, resulting in risk-assessment results with low accuracy. Second, qualitative risk assessment is a relatively macro-level prediction method. The research objects are mostly macro-level substances such as food additives and pollutants. It has difficulty performing the risk assessment of harmful microorganisms and thus has great limitations.

### 4.2. Quantitative Risk Assessment

In quantitative risk assessment, quantitative values are calculated by establishing a mathematical model based on quantitative data to represent the risk level and thereby assess the level of food safety risk. Quantitative risk-assessment methods mainly include BN, artificial neural networks (ANN), support vector machine (SVM), Monte Carlo (MC), and deep learning.

A BN is a kind of directed acyclic graph in which nodes can be used to represent variables. It uses graphs to represent a series of variables and their probability relationships. BN can associate information from different sources and distinguish uncertainty and variability; it is widely used in cold-chain logistics risk assessment [27,28]. Zhao et al. [29] considered various problems, such as the fact that a BN built using historical data and expert suggestions has a certain subjectivity, many factors lead to safety accidents in cold-chain logistics distribution systems, and the causes are complex and fuzzy. Accordingly, they adopted a scoring and search method to optimize the grid structure, used fuzzy set theory to determine conditional probability, and ran Bayesian reasoning to obtain the posterior probability of fresh logistics risk factors. On that basis, they obtained the risk impact value and risk-assessment results for risk factors in the failure of fresh food logistics. Liu [30], meanwhile, considered that a risk evaluation index system is similar to the idea of establishing a BN structure. Thus, the risk evaluation index system was transformed into a BN structure according to causal relationships, and a fuzzy BN was constructed to conduct forward causal reasoning and reverse diagnostic reasoning. On that basis, the risk level of the distribution system of Shuanghui Cold-Chain Logistics Company and factors prone to risk occurrence were evaluated. Chen et al. [31] suggested that a static BN cannot reflect the sequential transmission of risks between links in a cold chain. Thus, a dynamic BN was applied to a risk assessment model for dairy cold-chain logistics, and the probability and sensitivity of risks between links under a time series were calculated. Transport risk was found to be relatively high, and it was easy to transfer to the sales link with the change in time, while packaging process errors would have important effects on cold-chain risk.

An ANN is a mathematical model that simulates the behavior characteristics of animal neural networks for distributed and parallel information processing. It connects multiple processing units and achieves information processing by adjusting the connection relationships between a large number of internal nodes. A cold chain has a nonlinear trend; the risk factors and links often interact with each other, and the relationship is complex. ANN fully considers interactions between variables in the modeling process and can identify complex nonlinear relationships between variables. Therefore, using ANN for the risk assessment of cold-chain logistics has better accuracy [32,33]. Xu et al. [34] developed enterprise projects as evaluation indicators and improved the standard BP algorithm by adding momentum items and adjusting the learning rate, hidden-layer design, error function, and transformation function to predict risks in cold-chain logistics. Chen [35] selected indicators according to their importance and relevance, determined the risk indicator evaluation system, used principal component analysis to reduce the dimensionality of data for the cold-chain logistics of meat products, extracted eight principal components, and finally built a three-layer BP neural network to predict and evaluate the quality of risk of the output value. Wang et al. [36] used the Dempster–Shafer theory to optimize the expert evaluation method and built a radial basis function (RBF) neural network to evaluate risk in cold-chain marine logistics projects, focusing on the disadvantages of the long time and high risk of fresh agricultural products in cold-chain marine logistics. The results showed that the RBF neural network was superior to BP networks for logistics risk evaluation. ANN can not only evaluate risk in cold-chain logistics but also predict its demand. Xu et al. [37] used a wavelet neural network, BP neural network, genetic algorithm (GA) optimization neural network, particle swarm optimization neural network, and long short-term memory (LSTM) network to predict the demand for fresh agricultural products in Shandong Province, China. Although the prediction accuracy of optimized neural networks has been improved compared with traditional neural networks, the LSTM network has the highest prediction accuracy and is more suitable for demand prediction in cold-chain logistics. Currently, ANN-based risk assessment models for cold-chain logistics are mostly constructed based on traditional neural networks. With the continuous development of neural networks, deep neural networks with multiple hidden layers have been gradually applied to fresh sales forecasting [38,39], cold-chain environment forecasting [40,41], and other fields. The application of deep neural networks to cold-chain logistics risk assessment remains limited, and further research is needed.

SVM seeks the best compromise between the complexity and learning ability of a model based on limited sample information to obtain the best generalization ability. It has advantages for solving small-sample, nonlinear, high-dimensional pattern-recognition problems and has been widely used in early-warning security for cold-chain logistics [42,43]. Xu [44] used SVM to predict index data for fruit in cold-chain logistics and introduced cross-validation to screen the optimal penalty parameters and kernel functions to build an early-warning safety model for fruit cold-chain logistics. Yang et al. [45] built a quality and safety early-warning model for agricultural cold-chain logistics based on SVM and added an adaptive hybrid particle swarm optimization algorithm to optimize the SVM parameters to predict warning results. The results showed that the prediction effect of this method was better than that of traditional early-warning methods and had higher accuracy. Chang [46] used GA to optimize SVM parameters and build an optimal SVM model for the early-warning risk of different links in dairy cold-chain logistics, focusing on the large computational complexity of SVM and subjectivity in the selection of penalty function and kernel function models. The results showed that the prediction accuracy of an SVM model optimized by GA was better than that of a BP neural network algorithm and an SVM algorithm with default parameters.

The basic idea of the MC method is to estimate the probability of a random event based on the frequency of the event occurrence through experiments when the problem to be solved is the probability of the occurrence of a random event, and use this as a solution to the problem. It is mainly used to solve problems that cannot be solved using deterministic models. Therefore, it can solve the randomness and environmental uncertainty problems in the growth process of food microorganisms, and is widely used in research on microbial growth prediction, thereby conducting a risk assessment of microbial hazards. Bucur et al. [47] first used Baranyi and Roberts models to fit dynamic student length parameters. Then, MC simulation was used to predict the growth of Listeria monocytogenes on refrigerated ham at different temperatures, draw growth curves of Listeria monocytogenes at different temperatures, and evaluate the impact of temperature fluctuations on food risk. Huang [48] first used a one-step kinetic analysis method to determine the prediction model and kinetic parameters for the growth and survival of Salmonella in raw beef, obtaining prior knowledge. Then, Bayesian Markov chain Monte Carlo simulation was used to conduct a posterior analysis to predict the growth of Salmonella in cold-chain ground beef, which is helpful for assessing the risk of Salmonella in cold-chain storage of raw ground beef.

Deep learning uses data extracted by an algorithm to combine and transform low-level data features into complex high-level abstract features through a multilayer nonlinear neural network to complete the learning of complex tasks [49,50]. Shi et al. [51] used the tandem neuron structure in LSTM, which applies to the prediction of time series; converted lead content sampling data for fresh meat products 10 days before a major event into a risk level; and input it into the LSTM model to predict the lead content risk of frozen meat products the next day. Chen et al. [52] classified and graded the grey data of dairy products after pretreatment, obtained all levels of components through wavelet decomposition, and input them into an LSTM model to predict the risk level of all levels of components. Chen et al. [53] built a gated cycle unit network model to predict temperature changes in a cold-chain transport car body at different time series. Furthermore, the prediction accuracy was better than that of BP neural networks and cycle neural networks, thus overcoming the problems of the slow training rate of BP neural networks and the disappearance of the gradient of cycle neural networks. Wang et al. [54] constructed an early-warning risk model using a deep confidence network and multiclass fuzzy support vector machine. They extracted high-dimensional features from a large sample of grain data through the deep confidence network as the input for a multiclass fuzzy support vector machine to determine and rank of the risk level of hazards in the food supply chain.

In quantitative risk-assessment methods, BN applies to the influencing factors of risk within complex relationships. It can discover the influence relationships and causal effect strength between various risk factors, describe the degree of risk impact, and reverse track the source of risk events, thus achieving risk assessment and control. In a complex system, however, it is difficult to determine the interaction between nodes, and it is limited by prior knowledge. Thus, risk assessment is not applicable to complex systems. ANN has good fault tolerance for uncertain risk factors and nonlinear, fuzzy data in the cold chain. It has low requirements for data, thus reducing the difficulty of data processing, and it can assign qualitative indicators such as expert evaluation to the weight for correction. This reduces subjective participation and avoids the problem of low accuracy in qualitative risk assessment. SVM is suitable for dealing with problems such as a small amount of data, high dimensionality, and double classification. However, there are many links, regions, and influencing factors involved in a cold food chain, and the amount of sample data is huge, which will lead to low efficiency for SVMs, and food risk assessment is a multiclassification problem. It is often necessary, therefore, to construct multiple classifiers and combine optimization algorithms to improve prediction accuracy. The MC method is often used to explore the relationship between the intake of hazardous substances and risk, such as dietary exposure assessment. However, the relationship between variables in the MC method is one-way and there is no mutual influence, so it cannot reflect the impact of downstream information on upstream information, cannot determine the source of risk, and is difficult to achieve risk control. Deep learning requires a large amount of accurate data to ensure accurate prediction. However, cold food chain test data have a wide range of attributes, high complexity, and vulnerability to external influence. Therefore, data pretreatment is needed to improve the quality of sample data. Quantitative risk assessment is more objective than qualitative risk assessment and applies to risk assessment that requires high-accuracy results. However, quantitative risk assessment can only be studied on the basis of obtaining relevant data. Data accuracy directly affects the results of risk assessment; thus, it has certain limitations.

### 4.3. Comprehensive Qualitative and Quantitative Risk Assessment

Comprehensive qualitative and quantitative risk assessment is a comprehensive subjective and objective weighting evaluation method. An index system is constructed using the subjective weighting evaluation method, and a risk-prediction model is constructed using the objective weighting evaluation method to achieve accurate and effective risk assessment. Qualitative and quantitative risk-assessment methods are mainly divided into fuzzy comprehensive evaluation, AHP, fuzzy AHP, and the hidden Markov model (HMM).

The fuzzy comprehensive evaluation method is based on fuzzy mathematics and uses the principle of fuzzy relation synthesis to quantify factors that have unclear boundaries and are difficult to quantify; it also evaluates the subordination level of the evaluated object in terms of multiple factors [55,56]. Based on identifying the risk level of each influencing factor of risk in cold-chain logistics for fresh grapes, Wang [57] calculated the risk level of the cold chain using the fuzzy comprehensive evaluation method and concluded that the cold chain for fresh grapes was at the medium risk level. By determining the weight of each index using AHP, Xu et al. [58] calculated the comprehensive score for the logistics safety of fresh agricultural product sales in Hebei Province, China, using fuzzy comprehensive evaluation; overall logistics safety was found to be at the middle level. Guo [59] used principal component–grey correlation analysis to determine the key risk factors, used fuzzy comprehensive evaluation to determine the subordinate vector of the first-level indicators to the evaluation set, and used the maximum subordination principle to obtain the risk grade evaluation results. It was determined that the risk of the fresh aquatic product sales link was at the medium risk level as a whole, where quality, price, and service level risk were relatively high.

AHP refers to decomposing the whole system into different levels according to the interrelations, influence, and subordination among factors; the comparison of two factors at the same level; and the determination of the relative importance weight of the lowest-level index layer relative to the highest-level target layer as the basis for risk evaluation [60,61]. Qin et al. [62] used AHP to establish a risk evaluation index system for the cold-chain logistics of fresh agricultural products. They calculated and ranked the expert weights of the first- and second-level indicators based on a judgment matrix established by expert scoring and conducted a risk assessment on the four factors of environment, human resources, technology, and product processing. Zhang [63] suggested that traditional AHP is not applicable to situations where there is a lack of expert opinion and inconsistencies, and the expert weighting method is too subjective. Thus, the D number AHP and expert weight adaptive adjustment method were used to calculate index weight and expert weight, respectively, and the comprehensive risk value was obtained by multiplying the index scoring matrix to evaluate the risks faced by fresh cold-chain logistics enterprises. Padilla [64] used group AHP and the weighted average method to optimize the index weight and rank based on obtaining the expert weight and conducted a risk assessment of Mexico–China maritime cold-chain logistics. The results showed that management, infrastructure and technology, and physical operation processes were the three factors with the highest risk.

Fuzzy FAHP combines AHP with the fuzzy comprehensive evaluation method. First, index weight is calculated by the AHP method, and then the fuzzy comprehensive evaluation method is used for risk assessment and evaluation [65,66]. Xing [67] first used AHP to obtain the weights of risk measurement indicators; then used the fuzzy comprehensive evaluation method to conduct a three-level fuzzy evaluation from the indicator level to the target level, from the bottom to the top; and determined that the logistics risk of fresh e-commerce was at a medium level. Zhou [68] obtained a fuzzy comprehensive evaluation matrix by multiplying the index weight and the fuzzy evaluation matrix according to fuzzy theory and comprehensively evaluated a fresh-product cold-chain logistics supplier in the army based on the principle of maximum membership. Wu [69] built a risk-assessment index system based on an analysis of the uncertainty factors of cold-chain logistics and combined hierarchical analysis and fuzzy comprehensive evaluation to conduct a first- and second-level evaluation and comprehensive evaluation of the cold-chain logistics of meat in a logistics park. The first-level index and comprehensive cold-chain logistics risk were determined to be at the middle level.

HMM is a statistical model used to describe a Markov process with hidden unknown parameters. HMM consists of an observation process and a Markov chain, and can be described as a five element expression consisting of two state sets and three probability matrices. HMM has the advantages of simple modeling, a high detection rate, fast operation, and low computational complexity. It has been successfully applied to food risk assessment and prediction, food safety traceability, and other fields, with good evaluation accuracy. Liu et al. [70] first established an HMM based on DSSCAN clustering for the last link of the dairy product supply chain, obtained risk values under different time series, and achieved real-time risk level assessment. In addition, the HMM model was used to conduct probability analysis of the possible future safety levels of the system, and early warning risk analysis was conducted based on high indicator values, making up for the shortcomings of traditional static risk assessment. Aiming at the disadvantage that the Baum Welch algorithm in traditional HMM models can fall into local optimization, Lin et al. [71] proposed a hidden Markov model method based on the optimization of the cuckoo search algorithm. The initial HMM value is obtained through the global search function in the cuckoo search algorithm, and then the Baum Welch algorithm is used to modify the initial value, train the risk assessment model, and improve the accuracy of sterilized milk risk assessment results. Through the analysis of dynamic risk assessment results based on temperature characteristics, it is possible to predict future risk trends.

AHP takes the weight value of indicators calculated layer by layer as the basis for risk-grade judgment. It comprehensively considers the effect of indicators at all levels on food safety risk and reflects the hierarchical relevance of risk indicators at all levels. However, for a food field with a wide range of risk indicators, the correlation between indicators is complex, and the level and quantity of the judgment matrix and weights will increase, making calculation complicated; thus, it is not suitable for situations with too many risk indicators. Fuzzy comprehensive evaluation quantifies fuzzy and difficult-to-quantify risk indicators for risk assessment using fuzzy mathematics membership theory. However, when the index factor set is large, under the condition that the sum of the weight vectors is 1, the weight coefficient of the relative membership degree is often small, and the weight vector does not match the fuzzy judgment matrix, resulting in a super-fuzzy phenomenon. It is impossible to distinguish who has the higher membership degree, which affects the evaluation results. FAHP combines the advantages of fuzzy comprehensive evaluation and AHP, which can make the risk-assessment process more objective and make up for the shortcomings of AHP. Fuzzy theory is introduced to simplify the judgment matrix and calculation process. However, the above three methods are only used to assess the current risk status and cannot predict future changes in risk status based on historical and current data; thus, it is difficult to carry out the early warning of risk. HMM can not only evaluate current risks, but also achieve risk time series prediction, predict the development of risks in the future, and achieve dual evaluation of static and dynamic risks. Comprehensive qualitative and quantitative risk assessment can make full use of expert knowledge to achieve the transformation of risk indicators and grades from qualitative assessment to quantitative description. It not only avoids the disadvantages of low-accuracy assessment results caused by subjective factors such as expert experience but also does not need to collect a large amount of data to achieve risk quantification. It is more objective than qualitative risk assessment and has fewer limitations than quantitative risk assessment. However, its root is still expert knowledge, and the accuracy of evaluation results will be affected by subjective factors. In addition, such methods require a great deal of manual work to achieve the quantification of risk indicators and calculation of weights to determine the degree of risk, which is not conducive to daily risk assessment. Further, the risk-assessment process is too rough to handle risk assessment with high accuracy.

## 5. Problems and Challenges

As an important way to ensure cold-chain food safety, data-driven assessment has received considerable attention; however, it still faces various problems and challenges, which are summarized below.

(1)The data credibility of the cold food chain traceability system is low.

As an important part of cold-chain food safety risk prevention and control, the cold food chain traceability system generally has the problem of low data credibility. This is mainly attributable to the centralized structure of the cold food chain traceability system. Further, the core enterprises have absolute control over data entry, resulting in the possibility of data being tampered with, thus lowering data credibility. Blockchain technology is decentralized and tamper-proof, which means it can solve existing problems in the cold food chain traceability system. In the future, therefore, we should strengthen the integration of blockchain technology and food traceability systems to ensure that the data are authentic, effective, and tamper-proof, and aim to achieve the forward traceability and reverse traceability of cold food chains.

(2)The cold-chain food safety audit method has considerable limitations.

Cold-chain food safety auditing is an important part of cold-chain food safety management. Traditional cold-chain food safety auditing mainly focuses on on-site audits. The space span and irresistible factors have brought great challenges to auditing work, making traditional auditing methods more limited. Digital twinning technologies based on big data and artificial intelligence are constantly updating digital tools and information-transmission platforms, making it possible to conduct virtual cold-chain food safety audits. Digital twinning reflects space-, environment-, and quality-related information in the actual scene by building a virtual scene and integrating real food production data. A digital twin platform can achieve the virtual display of the production process, which is conducive to remote cold-chain food safety auditing; and it can save time and cost, and improve auditing efficiency. It represents a new approach to auditing.

(3)There is a lack of risk-assessment methods for nontraditional cold food chains.

Currently, the risk-assessment objects of cold food chains mainly focus on traditional food safety—that is, the risk assessment of physical, chemical, and biological hazards caused by pollution or the improper treatment of supply chain links (e.g., foreign matter, heavy metal pollution, and microbial hazards). The extensive application of risk-assessment methods has effectively controlled traditional food safety risks, but nontraditional food safety risks caused by human pollution and deliberate destruction are increasing, such as food additives exceeding standards, food counterfeiting, and human poisoning. The concealment, rapidity, diffusion, and unpredictability of nontraditional food safety have been significantly enhanced, while research on nontraditional food safety risk assessment is still in the initial stage. This increases the difficulty of prevention and control for production enterprises and government supervision. In the future, therefore, we should improve risk prevention and control theory in the nontraditional food safety field; excavate intelligent risk-assessment methods; strengthen information sharing among consumers, enterprises, and governments; and promote the formation of early-warning risk mechanisms to deal with food quality and safety risks in the new era.

## 6. Conclusions

Taking cold-chain food safety as the starting point, this study first constructs a keyword co-occurrence map and keyword clustering analysis knowledge map using CiteSpace and determines that data-driven analysis of risk assessment methods for cold food chain is a research hotspot in the field of risk assessment. Then, based on the data-driven perspective, the application of different data-driven methods to cold food chain risk assessment is elaborated on, and the advantages and limitations of each method are summarized. Finally, the problems and challenges of current cold food chain risk-assessment research are summarized, and suggestions are made in three aspects: the data credibility of cold food chain traceability systems, cold-chain food safety audit methods, and nontraditional cold food chain risk assessment. Because of their applicability, data-driven models have been widely used in the risk assessment of cold food chains, effectively improving the accuracy of risk assessment. Follow-up research can further strengthen the integration of food safety and intelligent information technology, excavate the intelligent evaluation means of cold food chain risk assessment, improve the scientific and practical nature of cold food chain risk assessment, and provide references for cold-chain food regulatory authorities to adopt risk prevention and control measures.

## Figures and Tables

**Figure 1 foods-12-01677-f001:**
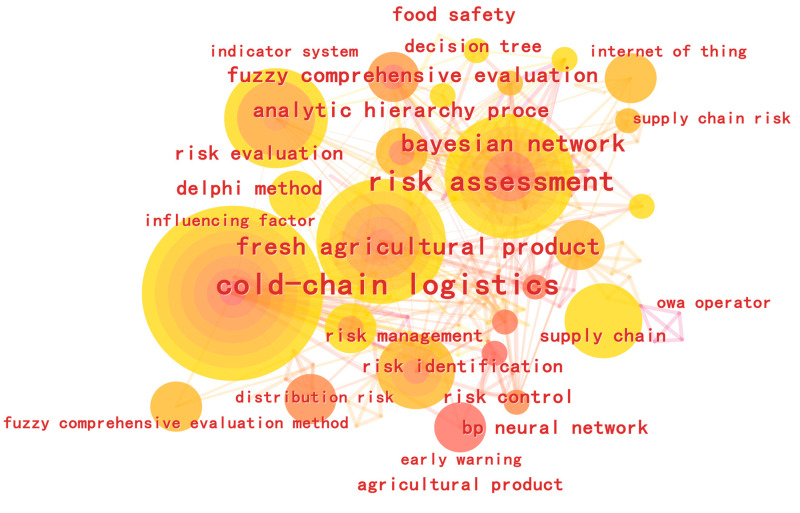
Keyword co-occurrence network.

**Figure 2 foods-12-01677-f002:**
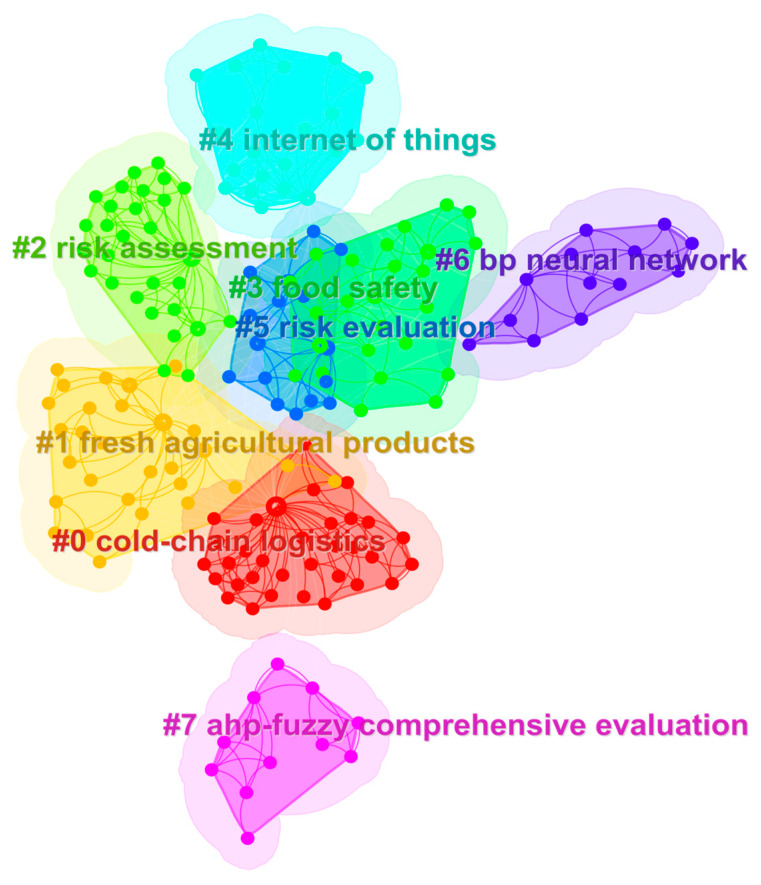
Knowledge map of keyword cluster analysis.

**Table 1 foods-12-01677-t001:** Research keywords and centrality statistics (top 20).

Sort	Frequency	Centrality	Keyword	First Occurrence Year
1	33	0.69	Cold-chain logistics	2012
2	24	0.52	Risk assessment	2011
3	16	0.18	Fresh agricultural product	2012
4	9	0.09	Analytic hierarchy process	2018
5	9	0.19	Bayesian network	2015
6	8	0.06	Fuzzy comprehensive evaluation	2013
7	8	0.06	Risk management	2014
8	7	0.04	Risk identification	2016
9	6	0.06	Supply chain	2012
10	5	0.06	Risk evaluation	2011
11	5	0.04	Risk control	2016
12	5	0.05	Food safety	2011
13	5	0.06	Decision tree	2011
14	4	0.04	Delphi method	2018
15	4	0.01	Influencing factor	2016
16	4	0.15	BP neural network	2014
17	3	0.03	Agricultural product	2016
18	3	0.03	Supply chain risk	2014
19	3	0.01	Fuzzy analytic hierarchy process	2016
20	3	0.00	Convolutional neural network	2022

## Data Availability

No new data were created or analyzed in this study. Data sharing is not applicable to this article.

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
