# Peer review of "Data-Driven Analysis of Risk-Assessment Methods for Cold Food Chains"

_foods, 2023, doi:10.3390/foods12081677_

Round 1

Reviewer 1 Report

This article focuses on food safety, a very important and cutting-edge topic in 2023. Specifically, it uses innovative CiteSpace software to identify the keywords of the past 18 years for the food cold chain. 

The article should be modified according to the following minor revisions:

- In Figure 2, it would be important to describe the clusters and argue them based on the reference years considered;

-In Table 1, explain why in your opinion those were the 20 keywords with high centrality always referring to their citation in the years;

-The conclusions need to be more argued because in this form they are very poor.

Otherwise, congratulations on the work performed.

Reviewer 2 Report

The research focus on an interesting topic, however, some limitations should be addressed.

There is no information provided on the selection of the 120 documents in the cold food chain safety field from 2006 to 2023 (l. 58-59). In particular, what kind of documents (academic, grey literature, etc.), where (which databases), and how (e.g., search terms) were selected? As all the remaining part of the paper relies on this, it is fundamental to explain in detail.

In Table 1, what does column "year" explain?

I don't see why the three main issues identified in Section 5 are highlighted.

In its present form, the section Conclusion is insufficient, providing almost no additional information. Also, I heavily missed a section that explains the importance of the results provided by the review.

Round 2

Reviewer 2 Report

The authors have addressed all of my concerns. A minor remark remained: I suggest shortly introducing China National Knowledge Internet, as readers outside of  China don't understand.
